# Action Inference by Maximising Evidence: Zero-Shot Imitation from Observation with World Models

**Xingyuan Zhang**[1,2*]**, Philip Becker-Ehmck**[1]**, Patrick van der Smagt**[1,3]**, Maximilian Karl**[1]

[1]Machine Learning Research Lab, Volkswagen Group, [2]Technical University of Munich,
[3]Eötvös Loránd University Budapest
{xingyuan.zhang,philip.becker-ehmck,maximilian.karl}@volkswagen.de

## Abstract

Unlike most reinforcement learning agents which require an unrealistic amount of environment interactions to learn a new behaviour, humans excel at learning quickly by merely observing and imitating others. This ability highly depends on the fact that humans have a model of their own embodiment that allows them to infer the most likely actions that led to the observed behaviour. In this paper, we propose Action Inference by Maximising Evidence (AIME) to replicate this behaviour using world models. AIME consists of two distinct phases. In the first phase, the agent learns a world model from its past experience to understand its own body by maximising the evidence lower bound (ELBO). While in the second phase, the agent is given some observation-only demonstrations of an expert performing a novel task and tries to imitate the expert's behaviour. AIME achieves this by defining a policy as an inference model and maximising the evidence of the demonstration under the policy and world model. Our method is "zero-shot" in the sense that it does not require further training for the world model or online interactions with the environment after given the demonstration. We empirically validate the zero-shot imitation performance of our method on the Walker and Cheetah embodiment of the DeepMind Control Suite and find it outperforms the state-of-the-art baselines. Code is available at: `https://github.com/argmax-ai/aime`.

## 1 Introduction

In recent years, deep reinforcement learning (DRL) has enabled intelligent decision-making agents to thrive in multiple fields [1, 2, 3, 4, 5, 6]. However, one of the biggest issues of DRL is sample inefficiency. The dominant framework in DRL is learning from scratch [7]. Thus, most algorithms require an incredible amount of interactions with the environment [1, 2, 3].

In contrast, cortical animals such as humans are able to quickly learn new tasks through just a few trial-and-error attempts, and can further accelerate their learning process by observing others. An important difference between biological learning and the DRL framework is that the former uses past experience for new tasks. When we try a novel task, we use previously learnt components and generalise to solve the new problem efficiently. This process is augmented by imitation learning [8], which allows us to replicate similar behaviours without direct observation of the underlying muscle movements. If the DRL agents could similarly harness observational data, such as the abundant online video data, the sample efficiency may be dramatically improved [9]. The goal of the problem is related to the traditional well-established Learning from Demonstration (LfD) field from the robotics community [10, 11], but instead of relying on knowledge from the engineers and researchers, e.g. mathematical model of robot's dynamic or primitives, we aim to let the robots learn by itself.

---

*Corresponding author.

37th Conference on Neural Information Processing Systems (NeurIPS 2023).

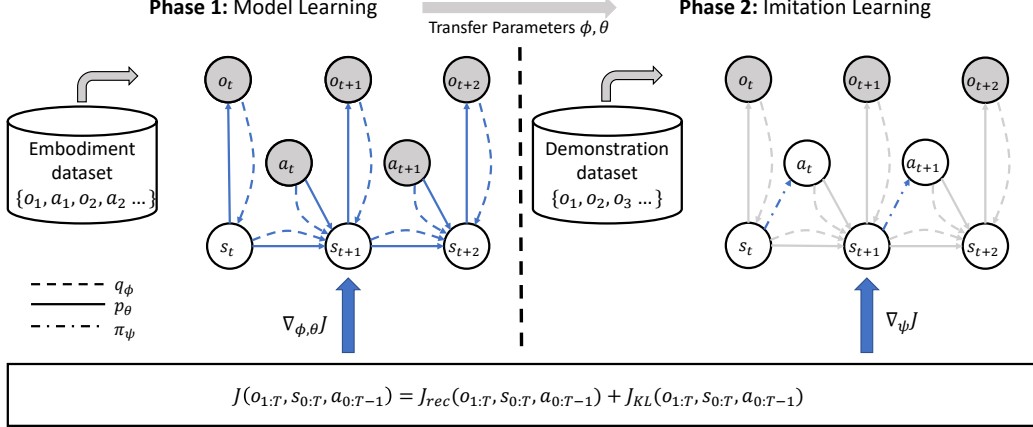

Figure 1: Overview of AIME algorithm. In phase 1, both observations and actions are provided by the embodiment dataset and the agent learns a variational world model to model the evidence of observations conditioned on the actions. Then the learnt model weights are frozen and transferred to phase 2. In phase 2, only the observations are provided by the demonstration dataset, so the agent needs to infer both states and actions. The action inference is achieved by the policy model which samples actions given a state. The grey lines indicate the world model parameters are frozen in phase 2. Both phases are optimised toward the same objective, i.e. the ELBO.

However, directly learning a model from observation-only sequences [12, 13] is insufficient for both biological and technical systems. Without knowing the actions that lead to the observations, the observation sequences are highly stochastic and multi-modal [14]. Trying to infer these unknown actions without prior knowledge of the world is difficult due to the problem of attributing which parts of the observations are influenced by the actions and which parts are governed by normal system evolution or noise.

Therefore, in this work, we hypothesise that in order to make best use of observation-only sequences, an agent has to first understand the notion of an action. This can be achieved by learning a model from an agent's past experiences where both the actions and their consequences, i.e. observations, are available. Given such a learnt model which includes a causal model of actions and their effects, it becomes feasible for an agent to infer an action sequence leading to given observation-only data.

In this work, we propose a novel algorithm, Action Inference by Maximising Evidence (AIME), to try to replicate the imitation ability of humans. The agent first learns a world model from its past experience by maximising the evidence of these experiences. After receiving some observation-only demonstrations of a novel task, the agent tries to *mimic* the demonstrator by finding an action sequence that makes the demonstration most likely under the learnt model. This procedure is shown in Figure 1.

Our contribution can be summarised as follows:

- We propose AIME, a novel method for imitation from observation. AIME first learns a world model by maximising the evidence of its past experience, then considers the policy as an action inference model and imitates by maximising the evidence of demonstration.
- We conduct experiments with a variety of datasets and tasks to demonstrate the superior performance of AIME compared with other state-of-the-art methods. The results showcase the zero-shot transferability of a learnt world model.

## 2 Problem formulation

Consider an MDP problem defined by the tuple $\{S, A, T, R\}$, where $S$ is the state space, $A$ is the action space, $T : S \times A \to S$ is the dynamic function and $R : S \to \mathbb{R}$ is the reward function. A POMDP adds partial observability upon an MDP with two components: the observation space $O$ and the emission function $\Omega : S \to O$. The six components of a POMDP can be categorised into three groups: $S$, $A$ and $T$ define the embodiment of our agent, $O$ and $\Omega$ define the sensors of our agent and

$R$ itself defines the task. The goal is to find a policy $\pi : S \to A$ which maximises the accumulated reward, i.e. $\sum_t r_t$.

In this paper, we want to study imitation learning within a fixed embodiment across different tasks. We presume the existence of two datasets for the same embodiment:

- Embodiment dataset $D_{\text{body}}$ contains trajectories $\{o_0, a_0, o_1, a_1 \dots\}$ that represent past experiences of interacting with the environment. This dataset provides information about the embodiment for the algorithm to learn a model. For example, in this paper, the dataset is a replay buffer filled while solving some tasks with the same embodiment. But in general, it may be any collection of past experiences of the embodiment.
- Demonstration dataset $D_{\text{demo}}$ contains a few expert trajectories $\{o_0, o_1, o_2 \dots\}$ of the embodiment solving a certain task defined by $R_{\text{demo}}$. The crucial difference between this dataset and the embodiment dataset is that the actions are not provided anymore since they are not observable from a third-person perspective.

The goal of our agent is to use information in $D_{\text{body}}$ to learn a policy $\pi$ from $D_{\text{demo}}$ which can solve the task defined by $R_{\text{demo}}$ as well as the expert who generated $D_{\text{demo}}$. For simplicity, we assume that the two datasets share the same observation space $O$ and the emission model $\Omega$.

## 3 Methodology

In this section, we describe our proposed method, AIME, in detail. AIME consists of two phases. In the first phase, the knowledge of the embodiment is learnt through a form of world model; while in the second phase, this knowledge is used to imitate the expert.

### 3.1 Phase 1: Model Learning

In the first phase, we need to learn a model to understand our embodiment. We achieve this by learning a world model. As an analogy to a language model, we define a world model as a probability distribution over sequences of observations. The model can be either unconditioned or conditioned on other factors such as previous observations or actions. For phase 1, the model needs to be the conditional distribution, i.e. $p(o_{1:T}|a_{0:T-1})$, to model the effect of the actions. When given an observation sequence, the likelihood of this sequence under the model is referred to as evidence.

In this paper, we consider variational world models where the observation is governed by a Markovian hidden state. In the literature, this type of model is also referred to as a state-space model (SSM) [15, 16, 17, 18, 19, 20]. Such a variational world model involves four components, namely

$$
\begin{aligned}
\text{encoder } & z_t = f_\phi(o_t), \\
\text{posterior } & s_t \sim q_\phi(s_t|s_{t-1}, a_{t-1}, z_t), \\
\text{prior } & s_t \sim p_\theta(s_t|s_{t-1}, a_{t-1}), \\
\text{decoder } & o_t \sim p_\theta(o_t|s_t).
\end{aligned}
$$

$f_\phi(o_t)$ is the encoder to extract the features from the observation; $q_\phi(s_t|s_{t-1}, a_{t-1}, z_t)$ and $p_\theta(s_t|s_{t-1}, a_{t-1})$ are the posterior and the prior of the latent state variable; while $p_\theta(o_t|s_t)$ is the decoder that decodes the observation distribution from the state. $\phi$ and $\theta$ represent the parameters of the inference model and the generative model respectively.

Typically, a variational world model is trained by maximising the ELBO which is a lower bound of the log-likelihood, or evidence, of the observation sequence, i.e. $\log p_\theta(o_{1:T}|a_{0:T-1})$. Given a sequence of observations, actions, and states, the objective function can be computed as

$$J(o_{1:T}, s_{0:T}, a_{0:T-1}) = J_{\text{rec}}(o_{1:T}, s_{0:T}, a_{0:T-1}) + J_{\text{KL}}(o_{1:T}, s_{0:T}, a_{0:T-1}), \tag{1}$$

$$\text{where } J_{\text{rec}}(o_{1:T}, s_{0:T}, a_{0:T-1}) = \sum_{t=1}^{T} \log p_\theta(o_t|s_t), \tag{2}$$

$$J_{\text{KL}}(o_{1:T}, s_{0:T}, a_{0:T-1}) = \sum_{t=1}^{T} -D_{\text{KL}}[q_\phi(s_t|s_{t-1}, a_{t-1}, f_\phi(o_t))||p_\theta(s_t|s_{t-1}, a_{t-1})]. \tag{3}$$

The objective function is composed of two terms: the first term $J_{\mathrm{rec}}$ is the likelihood of the observation under the inferred state, which is usually called the reconstruction loss; while the second term $J_{\mathrm{KL}}$ is the KL divergence between the posterior and the prior distributions of the latent state. To compute the objective function, we use the re-parameterisation trick [21, 22] to autoregressively sample the inferred states from the observation and action sequence.

Combining all these, we formally define the optimisation problem for this phase as

$$\phi^*, \theta^* = \underset{\phi, \theta}{\operatorname{argmax}} \, \mathbb{E}_{\{o_{1:T}, a_{0:T-1}\} \sim D_{\mathrm{body}}, s_{0:T} \sim q_\phi} [J(o_{1:T}, s_{0:T}, a_{0:T-1})]. \tag{4}$$

### 3.2 Phase 2: Imitation Learning

In the second phase, we want to utilise the knowledge of the world model from the first phase to imitate the expert behaviour from the demonstration dataset $D_{\mathrm{demo}}$ in which only sequences of observations but no actions are available. We derive our algorithm from two different perspectives.

**The Bayesian derivation** Since the actions are unknown in the demonstration, instead of modelling the conditional evidence in phase 1, we need to model the unconditional evidence, i.e. $\log p_\theta(o_{1:T})$. Thus, we also need to model the actions as latent variables together with the states. In this way, the reconstruction term $J_{\mathrm{rec}}$ will stay the same as eq. (2), while the KL term will be defined on the joint distribution of states and actions, i.e.

$$J_{\mathrm{KL}}(o_{1:T}, s_{0:T}, a_{0:T-1}) = \sum_{t=1}^{T} -D_{\mathrm{KL}}[q_{\phi,\psi}(s_t, a_{t-1}|s_{t-1}, f_\phi(o_t))||p_{\theta,\psi}(s_t, a_{t-1}|s_{t-1})]. \tag{5}$$

If we choose the action inference model in the form of a policy, i.e. $\pi_\psi(a_t|s_t)$, and share it in both posterior and prior, then the new posterior and prior can be factorised as

$$q_{\phi,\psi}(s_t, a_{t-1}|s_{t-1}, f_\phi(o_t)) = \pi_\psi(a_{t-1}|s_{t-1})q_\phi(s_t|s_{t-1}, a_{t-1}, f_\phi(o_t)) \tag{6}$$
$$\text{and } p_{\theta,\psi}(s_t, a_{t-1}|s_{t-1}) = \pi_\psi(a_{t-1}|s_{t-1})p_\theta(s_t|s_{t-1}, a_{t-1}) \tag{7}$$

respectively. When we plug them into the eq. (5), the policy term cancels and we will get a similar optimisation problem with phase 1 as

$$\psi^* = \underset{\psi}{\operatorname{argmax}} \, \mathbb{E}_{o_{1:T} \sim D_{\mathrm{demo}}, \{s_{0:T}, a_{0:T-1}\} \sim q_{\phi^*, \psi}} [J(o_{1:T}, s_{0:T}, a_{0:T-1})]. \tag{8}$$

The main difference between eq. (4) and eq. (8) is where the action sequence is coming from. In phase 1, the action sequence is coming from the embodiment dataset, while in phase 2, it is sampled from the policy instead since it is not available in the demonstration dataset.

**The control derivation** From another perspective, we can view phase 2 as a control problem. One crucial observation is that, as shown in eq. (1), given a trained world model, we can evaluate the lower bound of the evidence of any observation sequence given an associated action sequence as the condition. In a deterministic environment where the inverse dynamics model is injective, the true action sequence that leads to the observation sequence is the most likely under the true model. In general, the true action sequence may not necessarily be the most likely under the model. This is, however, a potential benefit of our approach. We are mainly interested in mimicking the expert's demonstration and may be better able to do so with a different action sequence.

Thus, for each observation sequence that we get from the demonstration dataset, finding the missing action sequence can be considered as a trajectory-tracking problem and can be tackled by planning. To be specific, we can find the missing action sequence by solving the optimisation problem

$$a^*_{0:T-1} = \underset{a_{0:T-1}}{\operatorname{argmax}} \, \mathbb{E}_{o_{1:T} \sim D_{\mathrm{demo}}, s_{0:T} \sim q_{\phi^*}} [J(o_{1:T}, s_{0:T}, a_{0:T-1})]. \tag{9}$$

If we solve the above optimisation problem for every sequence in the demonstration dataset, the problem will be converted to a normal imitation learning problem and can be solved with standard techniques such as behavioural cloning. We can also view this as forming an implicit inverse dynamics model (IDM) by inverting a forward model w.r.t. the actions.

To make it more efficient, we use amortised inference. We directly define a policy $\pi_\psi(a_t|s_t)$ under the latent state of the world model. By composing the learnt world model and the policy, we can form

**Algorithm 1:** AIME

**Data:** Embodiment dataset $D_{\text{body}}$, Demonstration dataset $D_{\text{demo}}$, Learning rate $\alpha$

*# Phase 1: Model Learning*
Initialise world model parameters $\phi$ and $\theta$
**while** *model has not converged* **do**
    $\{o_{1:T}, a_{0:T-1}\} \sim D_{body}$
    $s_0 \leftarrow 0$
    **for** $t = 1 : T$ **do**
        $s_t \sim q_\phi(s_t | s_{t-1}, a_{t-1}, f_\phi(o_t))$
    Compute objective function $J$ from eq. (1)
    Update model parameters $\phi \leftarrow \phi + \alpha \nabla_\phi J, \theta \leftarrow \theta + \alpha \nabla_\theta J$

*# Phase 2: Imitation Learning*
Initialise policy parameters $\psi$
**while** *policy has not converged* **do**
    $o_{1:T} \sim D_{demo}$
    $s_0 \leftarrow 0$
    **for** $t = 1 : T$ **do**
        $a_{t-1} \sim \pi_\psi(a_{t-1} | s_{t-1})$
        $s_t \sim q_\phi(s_t | s_{t-1}, a_{t-1}, f_\phi(o_t))$
    Compute objective function $J$ from eq. (1)
    Update policy parameters $\psi \leftarrow \psi + \alpha \nabla_\psi J$

a new generative model of the state sequence by the chain of $s_t \rightarrow a_t \rightarrow s_{t+1} \rightarrow a_{t+1} \ldots \rightarrow s_T$. Then we will get the same optimisation problem as eq. (8).

To sum up, in AIME, we use the same objective function – the ELBO – in both phases with the only difference being the source of the action sequence. We provide the pseudo-code for the algorithm in Algorithm 1 with the colour highlighting the different origins of the actions between the two phases.

## 4 Experiments

To test our method, we need multiple environments sharing an embodiment while posing different tasks. Therefore, we consider Walker and Cheetah embodiment from the DeepMind Control Suite (DMC Suite) [23]. Officially, the Walker embodiment has three tasks: stand, walk and run. While the Cheetah embodiment only has one task, run, we add three new tasks, namely run backwards, flip and flip backwards, inspired by previous work [24]. Following the common practice in the benchmark [19], we repeat every action two times when interacting with the environment. For both embodiments, the true state includes both the position and the velocity of each joint and the centre of mass of the body. In order to study the influence of different observation modalities, we consider three settings for each environment: *MDP* uses the true state as the observation; *Visual* uses images as the observation; *LPOMDP* uses only the position part of the state as the observation, so that information-wise it is identical to the *Visual* setting but the information is densely represented in a low-dimensional form.

To generate the embodiment and demonstration datasets, we train a Dreamer [19] agent in the Visual setting for each of the tasks for 1M environment steps. We take the replay buffer of these trained agents as the embodiment datasets $D_{\text{body}}$, which contain 1000 trajectories, and consider the converged policy as the expert to collect another 1000 trajectories as the demonstration dataset $D_{\text{demo}}$. We only use 100 trajectories for the main experiments, and the remaining trajectories are used for an ablation study. The performance of the policy is measured by accumulated reward. The exact performance of the demonstration dataset can be found in Appendix D. Besides the above embodiment datasets, we also study two datasets generated by purely exploratory behaviour. First, we use a random policy that samples uniformly from the action space to collect 1000 trajectories, and we call this the *random* dataset. Second, we train a Plan2Explore [24] agent for 1000 trajectories and label its replay buffer as the *p2e* dataset. Moreover, for the Walker embodiment, we also merge all the above datasets except the *run* dataset to form a *mix* dataset. This resembles a practical setting where one possesses a lot of experience with one embodiment and uses all of it to train a single foundational world model.

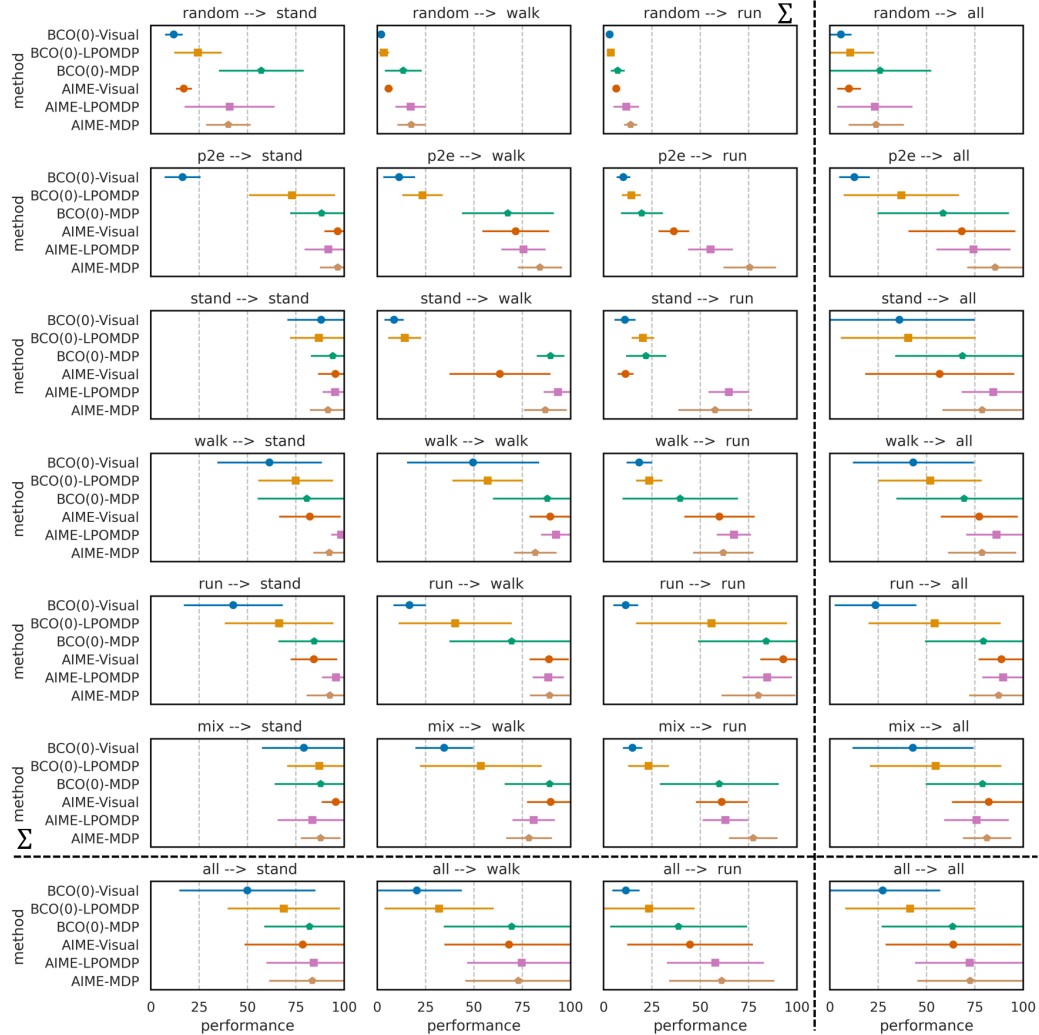

Figure 2: Performances on Walker. Each column indicates one task and its associated demonstration dataset, while each row indicates the embodiment datasets used to train the model. The title of each figure is named according to $D_{\mathrm{body}} \to D_{\mathrm{demo}}$. Numbers are computed by averaging among 100 trials and then normalised to the percentage of the expert's performance. Error bars are showing one standard deviation. The last row and column are averaged over the corresponding task or dataset. The error bar is large for them due to aggregating performance distributed in a large range.

## 4.1 Benchmark results

We mainly compare our method with BCO(0) [25]. BCO(0) first trains an IDM from the embodiment dataset and then used the trained IDM to label the demonstration dataset and then uses Behavioural Cloning (BC) to recover the policy. We do not compare with other methods since they either require further environment interactions [26, 27] or use a goal-conditional setting [28] which does not suit the locomotion tasks. More details about related works can be found in Section 5. The implementation details can be found in Appendix B.

The main results of our comparison are shown in Figure 2 and Figure 3. Overall, we can see that AIME largely outperforms BCO(0) in all the environment settings on Walker and on POMDP settings on Cheetah. AIME typically achieves the lowest performance on the Visual setting, but even that is comparable with BCO(0)-MDP which can access the true states. We attribute the good performance of AIME to two reasons. First, the world model has a better data utilisation rate than the IDM because the world model is trained to reconstruct whole observation sequences, while the IDM only takes

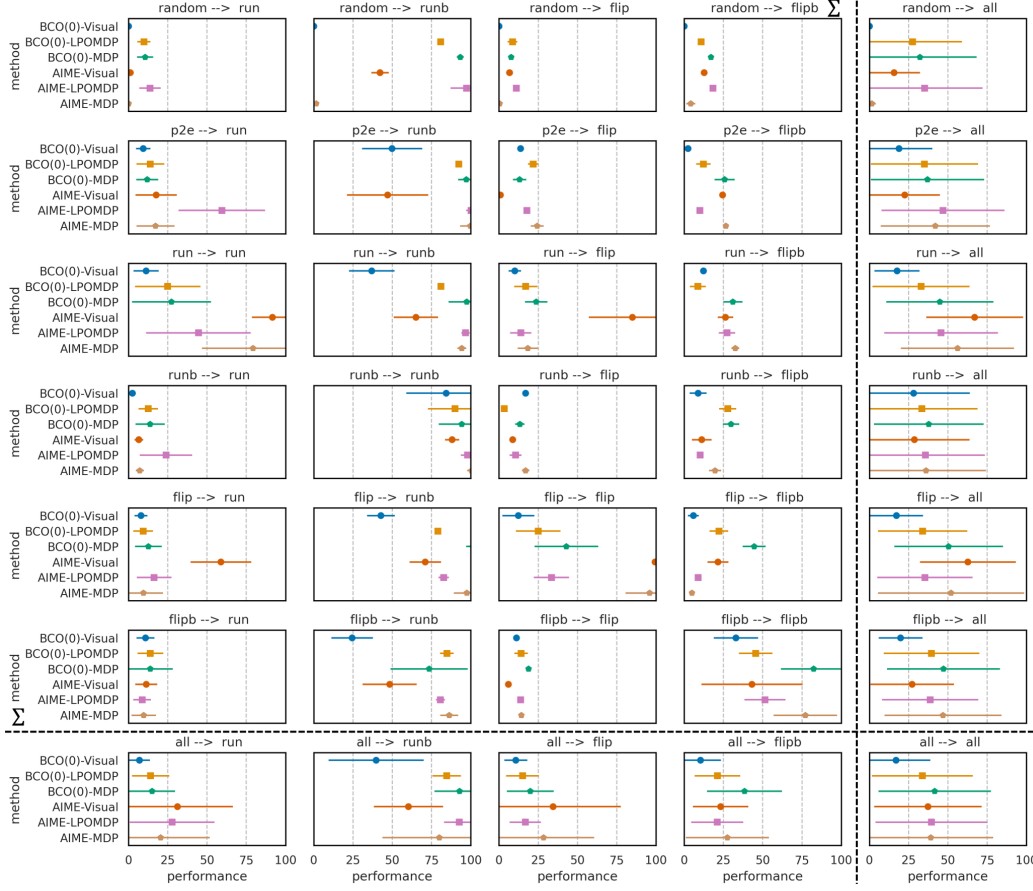

Figure 3: Performances on Cheetah. Each column indicates one task and its associated demonstration dataset, while each row indicates the embodiment datasets used to train the model. The title of each figure is named according to $D_{\text{body}} \rightarrow D_{\text{demo}}$. *runb* and *flipb* are short hands for *run backwards* and *flip backwards*. Numbers are computed by averaging among 100 trials and then normalised to the percentage of the expert's performance. Error bars are showing one standard deviation. The last row and column are averaged over the corresponding task or dataset. The error bar is large for them due to aggregating performance distributed in a large range.

short clips of the sequence and only predicts the actions. Thus, the world model has less chance to overfit, learns better representations and provides better generalisation. Second, by maximising the evidence, our method strives to find an action sequence that leads to the same outcome, not to recover the true actions. For many systems, the dynamics are not fully invertible. For example, if a human applies force to the wall, since the wall does not move, one cannot tell how much force is applied by visual observation. The same situation applies to the Walker and Cheetah when certain joints are locked due to the singular pose. This same phenomenon is also discussed in [28].

We also find that, comparing with the Walker experiments, the performance on Cheetah is lower and the improvement offered by AIME is smaller. We think it is because the setup for Cheetah is much harder than Walker. Although the tasks sound similar from the names, e.g. flip and flip backward, due to the asymmetrical structure of the embodiment, the behaviour for solving the tasks can be quite different. The difference limits the amount of knowledge that can be transferred from the embodiment dataset to the demonstrations. Moreover, some tasks are built to be hard for imitation. For example, in the demonstration of the flip tasks, the cheetah is "flying" in the air and the actions taken there is not relevant for solving the tasks. That leaves only a few actions in the sequence that are actually essential for solving the task. We think this is more challenging for AIME since it needs to infer a sequence of actions, while BCO(0) is operating on point estimation. That is, when the first few actions cannot output reasonable actions to start the flip, then the later actions will create a very noisy

gradient since none of them can explain the "flying". In general, poorly modelled regions of the world may lead to noisy gradients for the time steps before it. On the other hand, we can also find most variants achieve a good performance on the run backward demonstration dataset, which is mainly due to low expert performance (see Appendix D) for the task that makes imitation easy. Last but not least, since we follow the common practise for the benchmark [19], the Cheetah embodiment is operated on 50Hz which is much higher than the 20Hz used in Walker. Higher frequency of operation makes the effect of each individual action, i.e. change in the observation, more subtle and harder to distinguish, which poses an additional challenge for the algorithms.

**Influence of different datasets** As expected, for almost all the variants of methods, transferring within the same task is better than transferring between different tasks. In these settings, BCO(0)-MDP is comparable with AIME. However, AIME shines in cross-task transfer. Especially when transferring between run and walk tasks and transferring from stand to run on Walker, AIME outperforms the baselines by a large margin, which indicates the strong generalisability of a forward model over an inverse model. We also find that AIME makes substantially better use of exploratory data. On Walker, AIME largely outperforms baselines when using the p2e dataset as the embodiment dataset and outperforms most variants when using the random dataset as the embodiment dataset. Moreover, when transferring from the mix dataset, except for the MDP version, AIME outperforms other variants that train the world model on just any single individual task dataset of the mixed dataset. This showcases the scalability of a world model to be trained on a diverse set of experiences, which could be more valuable in real-world scenarios.

**Influence of observation modality** Compared with BCO(0), AIME is quite robust to the choice of observation modality. We can see a clear ladder pattern with BCO(0) when changing the setting from hard to easy, while for AIME the result is similar for each modality. However, we can still notice a small difference when comparing LPOMDP and Visual settings. Although these observations provide the same information, we find AIME in the LPOMDP setting performs better than in the Visual setting in most test cases. We attribute it to the fact that low-dimension signals have denser information and offer a smoother landscape in the evidence space than the pixels so that it can provide a more useful gradient to guide the action inference. Surprisingly, although having access to more information, AIME-MDP performs worse than AIME-LPOMDP on average. The biggest gaps happen when transferring from exploratory datasets, i.e. the p2e dataset on Walker and the random dataset on Cheetah. We conjecture this to the fact the world model is not trained well with the default hyper-parameters, but we defer further investigation to future work.

## 4.2 Ablation studies

In this section, we conduct some ablation studies to investigate how AIME's performance is influenced by different components and design choices. We will mainly focus on using the *mix* embodiment dataset and transfer to *run* task, which represents a more realistic setting where we want to use experience from multiple tasks to transfer to a new task.

**Sample efficiency and scalability** To test these properties, we vary the number of demonstrations within $\{1, 2, 5, 10, 20, 50, 100, 200, 500, 1000\}$. We also include BC with the true action as an oracle baseline. The results are shown in Figure 4. BCO(0) struggles with low-data scenarios and typically needs at least 10 to 20 demonstrations to surpass the performance of a random policy. In contrast, AIME demonstrates continual improvement with as few as 2 trajectories. And surprisingly, thanks to the generalisation ability of the world model, AIME even outperforms oracle BC when the demonstrations are limited. These demonstrate the superior sample efficiency of the method. Moreover, the performance of AIME keeps increasing as more trajectories are provided beyond 100, which showcases the scalability of the method.

**Objective function** The objective function, i.e. ELBO, consists of two terms, the reconstruction term $J_{\text{rec}}$ and the KL term $J_{\text{KL}}$. To investigate the role that each term plays in AIME, we retrain two variants of AIME by removing either of the terms. As we can see from Figure 5, removing either term will negatively impact the results. When we compare the two variants, only using the KL term is better in settings with low-dimensional signals, while using only the reconstruction term yields a slightly better result for the high-dimensional image signal. But on all settings, the performance of using only the KL term is very close to the one that use both terms. This suggests that the latent state in the world model has already mostly captured the essential part of the environment. Although it is

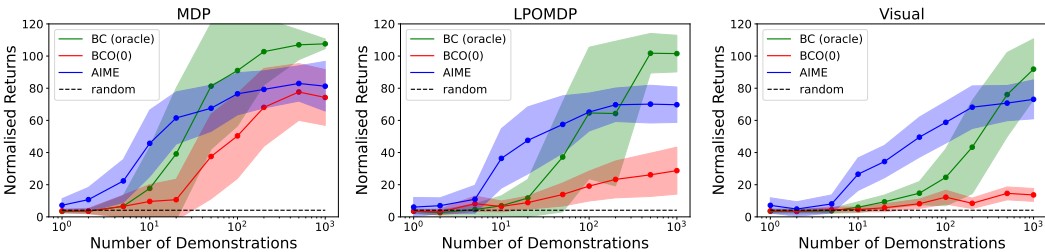

Figure 4: Ablation of the number of demonstrations on *mix → run* transfer on the Walker embodiment. The performance is shown as the normalised returns over 3 seeds and 100 trials for each seed. The shaded region represents one standard deviation.

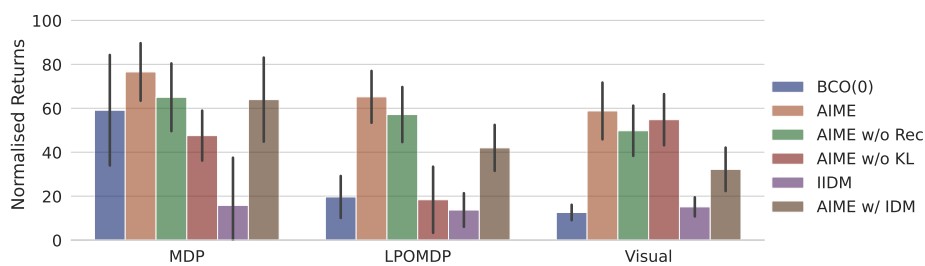

Figure 5: Ablation studies on *mix → run* transfer on the Walker embodiment. Numbers are computed by averaging among 3 seeds and 100 trials for each seed, and then normalised to the percentage of the expert's performance. Error bars are showing one standard deviation.

still worse than using both terms, it sheds some light on the potential of incorporating decoder-free models [29] into the AIME framework.

**Components** Compared with the BCO(0) baseline, AIME consists of two distinct modifications: one is to use an SSM to integrate sequences and train the policy upon its latent representation; the other is to form an implicit IDM via gradients rather than training an IDM explicitly. We design two baselines to investigate the two components. First, to remove the SSM, we train a forward dynamics model directly on the observations of the embodiment dataset and use that as an implicit IDM for imitation on the demonstration dataset. We term this variant IIDM. Second, we train a separate IDM upon the trained latent state of the world model and use that to guide the policy learning in phase 2. The detailed derivation of the IDM formulation can be found in Appendix C. Figure 5 clearly demonstrates the significance of the latent representation for performance. Without the latent representation, the results are severely compromised across all settings. However, when compared to BCO(0), the IIDM does provide assistance in the high-dimensional Visual setting, where training an IDM directly from the observation space can be extremely challenging. While having IDM on the latent representation leads to a better performance comparing with BCO(0), but it still performs worse than AIME, especially on the POMDP settings.

## 5 Related works

**Imitation learning from observations** Most previous works on imitation learning from only observation can be roughly categorised into two groups, one based on IDMs [25, 9, 30, 28] and one based on generative adversarial imitation learning (GAIL) [31, 26, 27]. The core component of the first group is to learn an IDM that maps a state transition pair to the action that caused the transition. [25, 9] use the IDM to label the expert's observation sequences, then solve the imitation learning problem with standard BC. [30, 28] extend the IDM to a goal-conditioned setting in which the IDM is trained to be conditioned on a future frame as the goal instead of only the next frame. During deployment, the task is communicated on the fly by the user in the form of key-frames as goals. The

setup mainly suits for the robot manipulation tasks in their paper since the user can easily specify the goals by doing the manipulation himself, but not suits for the locomotion tasks, in which it is not clear what a long-term goal of observation is and also not practical set the next observation as the goal and demonstrate that in a high frequency by the user. Different from these methods, our approach uses a forward model to capture the knowledge of the embodiment. In the second group of approaches, the core component is a discriminator that distinguishes the demonstrator's and the agent's observation trajectories. Then the discriminator serves as a reward function, and the agent's policy is trained by RL [31]. As a drawback, in order to train this discriminator the agent has to constantly interact with the environment to produce negative samples. Different from these methods, our method does not require further interactions with the environment, enabling zero-shot imitation from the demonstration dataset. Besides the majority, there are also works [32, 33] don't strictly fit to the two groups. [32] also use forward model like us by learning a latent action policy and a forward dynamic based on the latent action. However, it still needs online environment interactions to calibrate the latent actions to the real actions. [33] is hybrid method that uses both of the components and focus on a setting that the demonstrations are coming from a different embodiment.

**Reusing learnt components in decision-making** Although transferring pre-trained models has become a dominant approach in natural language processing (NLP) [34, 35, 36] and has been getting more popular in computer vision (CV) [37, 36], reusing learnt components is less studied in the field of decision-making [7]. Most existing works focus on transferring policies [38, 9, 7]. On the other hand, the world model, a type of powerful perception model, that is purely trained by self-supervised learning lies behind the recent progress of model-based reinforcement learning [39, 17, 19, 40, 41, 42, 43, 44]. However, the transferability of these world models is not well-studied. [24] learns a policy by using a pre-trained world model from exploration data and demonstrates superior zero-shot and few-shot abilities. We improve upon this direction by studying a different setting, i.e. imitation learning. In particular, we communicate the task to the model by observing the expert while [24] communicates the task by a ground truth reward function which is less accessible in a real-world setting.

## 6 Discussion & conclusion

In this paper, we present AIME, a model-based method for imitation from observations. The core of the method exploits the power of a pre-trained world model and inverses it w.r.t. action inputs by taking the gradients. On the Walker and Cheetah embodiments from the DMC Suite, we demonstrate superior performance compared to baselines, even when some baselines can access the true state. The results showcase the zero-shot ability of the learnt world model.

Although AIME performs well, there are still limitations. First, humans mostly observe others with vision. Although AIME works quite well in the *Visual* setting, there is still a gap compared with the LPOMDP setting where the low-dimensional signals are observed. We attribute this to the fact that the loss surface of the pixel reconstruction loss may not be smooth enough to allow the gradient method to find an equally good solution. Second, in this paper, we only study the simplest setting where both the embodiment and sensor layout are fixed across tasks. On the other hand, humans observe others in a third-person perspective and can also imitate animals whose body is not even similar to humans'. Relaxing these assumptions will open up possibilities to transfer across different embodiments and even directly from human videos. Third, for some tasks, even humans cannot achieve zero-shot imitation by only watching others. This may be due to the task's complexity or completely unfamiliar skills. So, even with proper instruction, humans still need to practise in the environment and learn something new to solve some tasks. This motivates an online learning phase 3 as an extension to our framework. We defer these topics to future work.

We hope this paper demonstrates the great potential of transferring a learnt world model, incentivises more people to work in this direction and encourages researchers to also share their learnt world model to contribute to the community.

## Acknowledgments and Disclosure of Funding

We want to acknowledge Elie Aljalbout for the insightful discussion during the initial stage of the project and Botond Cseke for mathematical support.

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

## A   Computational resources

On a GTX 1080Ti graphics card, AIME typically requires 10 hours of training for phase 1 and 5 hours of training for phase 2 with MDP and LPOMDP setups. The required time nearly doubled when running with Visual settings, due to heavier visual backbones and rendering. We conduct experiments on a shared local cluster which uses A100 and RTX8000 GPUs. The newer GPUs can slightly improve the training speeds but not much since the main computational bottleneck is the recurrent structure. In terms of one GTX 1080Ti, it will require roughly 50 GPU days to produce the benchmark results.

## B   Implementation and training details

We implement all listed methods in PyTorch [45].

For the world model, we use RSSM [17, 19], which offers state-of-the-art performances by splitting the latent state to be a combination of deterministic and stochastic components. The RSSM implementation is largely following Dreamer-v1 [19] with continuous stochastic and deterministic variables. Although newer versions of Dreamer [41, 42] offer some new tricks to improve performance, we initially choose not to use them for the sake of simplicity. We use a slightly larger state space for our experiment with 512 deterministic and 128 stochastic dimensions and find it generally eases the policy training process to collect the datasets. For the Visual setting, the encoder and decoder are implemented with CNNs. The decoder output a Gaussian distribution with the mean output by CNN and a fixed variance of 1. For the low-dimensional settings, the encoder is implemented as an identity function while the decoders are Gaussian distributions with both the mean and variance parameterised by MLPs. The deterministic part of the state is implemented as a GRU cell [46]. For the default hyper-parameters, we do not use any free nats [19], KL scaling [19] and KL balancing [41] tricks in the literature to relax the constraint of the KL term. When decoding low-dimensional signals, we sometimes observed the decoder yielding a degenerate solution as found in [47]. We use their $\beta$-nll to remedy this problem, and since it re-weights the reconstruction term, we re-weight the KL term accordingly to maintain the balance.

Without further mention, all the CNN encoders and decoders above are implemented as in [39, 17], while all the MLPs are with 2 hidden layers and 128 units of each layer with ELU [48] activation function. All the components are trained with Adam optimiser with a learning rate of $1e{-}3$. For the stochastic policy, the output distribution is modelled by a TanhGaussian distribution [49] with both the mean and variance parameterised by neural networks.

For AIME, we consider 100 gradient steps as an epoch. For phase 1, the model is trained for 1000 epochs, while for phase 2 we train the policy for 500 epochs. Both the final model and policy are from the last epoch without any early stopping criteria.

When training the world model on the Cheetah dataset, we find the default hyper-parameters cannot stably train a good world model. Thus, we adapt the implementation to exactly the same network structure as the origin repository. Specifically, the decoders of low-dimensional observations are also with a fixed variance of 1 and all the MLPs are widened to 512 neurons in each hidden layer and equipped with Layer Normalisation [50]. For the hyper-parameters, the learning rate is decreased to $3e{-}4$, and we use free nats of 1.0 and KL balancing of 0.8 to mitigate the collapse and unstable problem of the KL term. For the LPOMDP setting, we also set the KL scaling parameter $\beta = 0.0002$ to relax the constraint. One thing that needs to mention is, while the tricks about the KL term are helpful for model training, they hurt the results in phase 2. It could be because, in phase 2, the model is frozen, so that no-more stability issues will be encountered. So in this case it is better to optimise the policy with the true ELBO.

To be strict with our setup of the two phases, we retrain the world model after data collection for all the experiments. However, one can also directly use the world model from the trained dreamer agent. We empirically find these models yield similar results with the world model retrained afterwards on the same reply buffer. One caveat is that, although it is tempting to also reuse the trained policy as initialisation in phase 2, we found it is actually harmful to the performance. We conjecture that it is due to learnt policies being stuck in some local minima that they are unable to escape.

For the BCO(0) baseline, the IDM and policy are built by using the same network architecture with the world model to make a fair comparison. The observations are first processed by the encoder network, and then get stacked to deal with the temporal information. An MLP is used to decode the stacked representation to the output distribution. We stack 5 consecutive in this work. We did a grid search about the width, depth of the MLP and also the number of stacking frames and didn't find any increase of the performance. Following the original paper, we split the datasets by $7 : 3$, and choose the finial model based on the validation loss.

## C   AIME with IDM

In this section, we introduce an alternative variant of AIME which also uses IDM. Recall that in the Bayesian derivation, we factorised both the posterior and prior of the joint distribution of state and action with a shared policy network, as in eq. (6) and eq. (7). Alternatively, we can re-factorise the posterior with IDM, that is

$$q_{\phi,\theta}(s_t, a_{t-1}|s_{t-1}, f_\phi(o_t)) = q_\phi(a_{t-1}|s_{t-1}, f_\phi(o_t))q_\phi(s_t|s_{t-1}, a_{t-1}, f_\phi(o_t)). \qquad (10)$$

One thing that needs to be noticed here is that the IDM is not in the familiar form of $q_\phi(a_{t-1}|s_{t-1}, s_t)$. This is because the latent state in the world model is action dependent so the familiar form is non-casual in the world model. But we should highlight here that this non-casual structure is a result of the model we used in phase 1 since we want to reuse the knowledge learnt there. For example, one can also factorise the joint posterior as

$$q_\phi(s_t, a_{t-1}|s_{t-1}, f_\phi(o_t)) = q_\phi(a_{t-1}|s_{t-1}, s_t)q_\phi(s_t|s_{t-1}, f_\phi(o_t)). \qquad (11)$$

However, in this case, the model is different for phase 1. In this section, we stick to using the factorisation in eq. (10). Since a new IDM component is introduced, the objective of both phase 1 and phase 2 need to be modified. For phase 1, since actions are available in the dataset, the IDM can be treated as a decoder and trained by maximising likelihood. That is, we add $J_{\text{IDM}}^{\text{p1}}(\tau^{(T)}) = \sum_{t=0}^{T-1} \log q_\phi(a_t|s_t, f_\phi(o_{t+1}))$ to the objective function. For phase 2, since actions are not available, the IDM serves as the posterior and guides the prior policy through a KL divergence, i.e. $J_{\text{IDM}}^{\text{p2}}(\tau^{(T)}) = \sum_{t=0}^{T-1} -D_{KL}[q_\phi(a_t|s_t, f_\phi(o_{t+1}))||\pi_\psi(a_t|s_t)]$.

One caveat about this formulation is that, in phase 1, the IDM forms a loop on the graphical model. In order to stabilise the training process, we detach the gradient from the IDM to the rest of the network.

## D   Dataset details

Here we provide extra information about the datasets. The expert return which we normalised against is shown in Table 1 and Table 2.

Table 1: Average expert return of each demonstration dataset of Walker.

| $D_{\text{demo}}$ | Average return |
| --- | --- |
| stand | 957.87 (max: 1000) |
| walk | 943.79 (max: 1000) |
| run | 604.10 (max: 1000) |

Table 2: Average expert return of each demonstration dataset of Cheetah.

| $D_{\text{demo}}$ | Average return |
| --- | --- |
| run | 888.65 (max: 1000) |
| run backwards | 218.50 (max: 500) |
| flip | 485.79 (max: 500) |
| flip backwards | 379.91 (max: 500) |

Table 3: Result on V-D4RL main datasets. Embodiment datasets are marked on the left, and the demonstration datasets are chosen to be the expert dataset for each task in the origin environment. Values are averaged over 100 trajectories and reported as accumulated reward divided by 10, as suggested in the V-D4RL paper.

| | BCO(0) | AIME |
|---|---|---|
| walker-random | $2.11 \pm 0.91$ | $\mathbf{12.36 \pm 4.69}$ |
| walker-medium_replay | $6.54 \pm 6.56$ | $\mathbf{10.18 \pm 4.33}$ |
| walker-mix | $5.32 \pm 5.23$ | $\mathbf{8.49 \pm 3.60}$ |
| cheetah-random | $0.01 \pm 0.01$ | $\mathbf{9.48 \pm 4.72}$ |
| cheetah-medium_replay | $15.47 \pm 7.38$ | $\mathbf{31.47 \pm 16.14}$ |
| cheetah-mix | $16.08 \pm 6.61$ | $\mathbf{40.27 \pm 11.52}$ |

# E  Experiments on V-D4RL datasets

We provide here some additional results of AIME on V-D4RL datasets [51] to showcase that AIME can also work with datasets collected by non-model-based methods. V-D4RL provides multiple different datasets for the Walker and Cheetah embodiments from DMC Suite, and it is original designed for offline RL with visual inputs. The datasets are collected by running a few model-free RL methods and either keep the replay buffer or rollout from a policy checkpoints. Since our setting requires a bit more exploration in the embodiment datasets to understand the embodiment, we choose to use their random and medium_replay datasets as the embodiment datasets. The expert datasets are used as the demonstration datasets. Same as what we did for the Walker embodiment in the main text, we also mix the two embodiment datasets for each embodiment to form a mix dataset.

The results on these datasets are shown in Table 3. We can see that the performance of both BCO(0) and AIME is generally low, but AIME still outperform BCO(0) which proves AIME can also handle datasets generated by model-free methods. The low performance is due to a more constrained setup of the task, i.e. less amount of embodiment data and less diversity. Except the cheetah-medium_replay having $400$ trajectories, the other three datasets provided by V-D4RL have only $200$ trajectories, which is much less than the $1000$ trajectories in the main experiments. Moreover, it is already shown from Figure 2 and Figure 3 that random datasets do not help much in learning a model, and intuitively the medium_replay dataset is better but still does not contain enough information to solve the task.

We also conduct experiments on the V-D4RL distracting datasets, to test the performance of AIME on distracting datasets. For the Walker embodiment, the benchmark provides random datasets with a distraction level of easy, medium, and hard. We also merge these three levels to form a mix dataset. Moreover, we also merge this mix dataset with the mix dataset in the second experiment to form a total_mix dataset. We treat these five datasets as the embodiment dataset and the expert dataset as the demonstration dataset. For the Cheetah embodiment, the benchmark provides medium and expert datasets with a distraction level of easy, medium, and hard. We subsample the medium datasets to get $200$ trajectories from each level, then merge that with the mix dataset in the second experiment to form a total_mix dataset. Then the algorithms are using this total_mix dataset as the embodiment dataset and the expert dataset as the demonstration dataset.

As we can see from the result from Table 4, although we still outperform the BCO(0) baseline, AIME is impacted significantly by the distractions. This behaviour is expected since the world model is trained with reconstruction loss. It is not easy to handle observations with distractions. A potential solution to this problem is to freeze only the dynamics part of the world model and allowing encoders and decoders to fine-tune their parameters in the second phase. We leave these improvements for our future works.

# F  Additional plots

In this section, we will present some additional plots to complement the main text and provide further insights.

Table 4: Result on V-D4RL distracting datasets. Embodiment datasets are marked on the left, and the demonstration datasets are chosen to be the expert dataset for each task in the origin environment. Values are averaged over 100 trajectories and reported as accumulated reward divided by 10, as suggested in the V-D4RL paper.

|  | BCO(0) | AIME |
|---|---|---|
| walker-easy | $2.10 \pm 0.88$ | $\mathbf{4.73 \pm 2.54}$ |
| walker-medium | $2.15 \pm 0.87$ | $\mathbf{3.94 \pm 0.99}$ |
| walker-hard | $2.15 \pm 0.97$ | $\mathbf{4.16 \pm 1.98}$ |
| walker-mix | $2.12 \pm 0.86$ | $\mathbf{3.81 \pm 2.07}$ |
| walker-total_mix | $2.15 \pm 0.71$ | $\mathbf{12.66 \pm 4.51}$ |
| cheetah-total_mix | $16.61 \pm 7.28$ | $\mathbf{32.40 \pm 14.52}$ |

Additional to Figure 2 and Figure 3, we also provide detailed profile plots in Figure 6 and Figure 7 as recommended in [52]. We can see that AIME is normally more stable w.r.t. the performance by having a smaller decay region. It is clearly shown on such tasks as *walk → walk* and *run → run* on Walker where BCO(0)-MDP has some trails with very low performance, while all variants of AIME maintain decent performance.

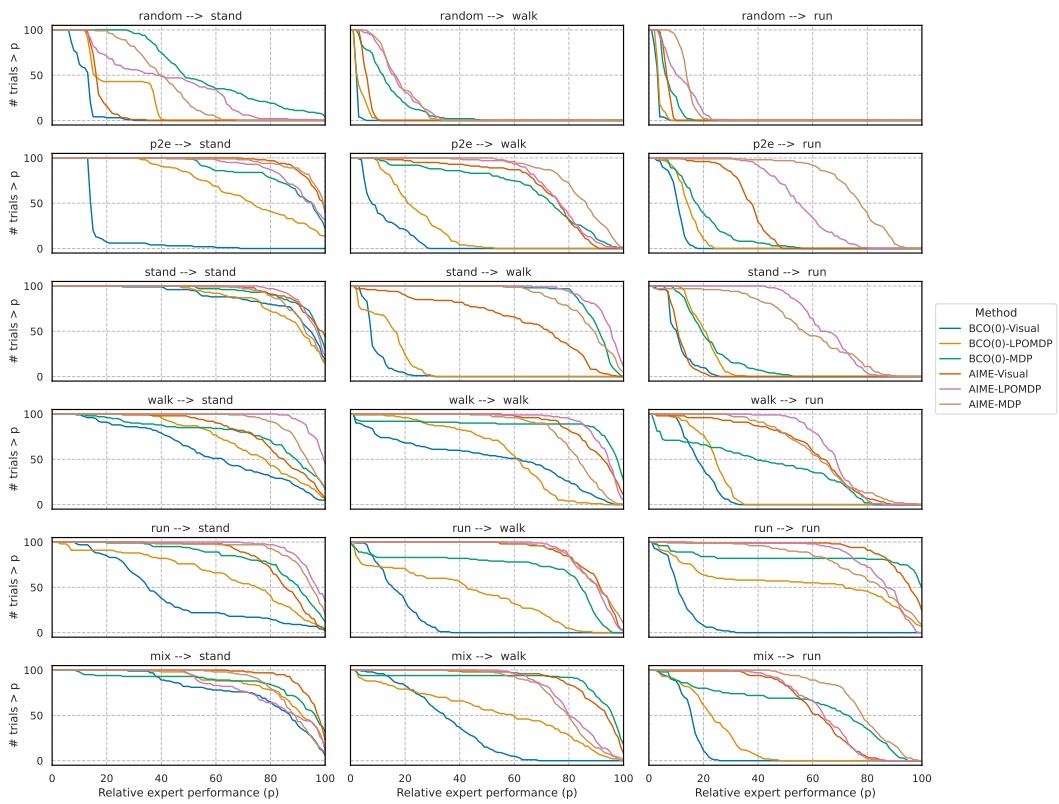

Figure 6: Performance distributions of each method on Walker tasks.

We also present some representative training curves of AIME's phase 2 from our experiments in Figure 8. The first three figures show the transfer from the mix dataset to the run task in the three settings which are the typical success cases of AIME. During the course of training, ELBO is maximised towards convergence and the MSE between the generated actions and the true actions decreases. We can also see that for the MDP and LPOMDP settings, the converged ELBO is lower than the ELBO when evaluated with the true action sequence, indicating there is still space for improvement. However, for the Visual setting, the converged ELBO exceeds the one with true actions, which should be attributed to the over-fitting of the world model from phase 1. The last three figures

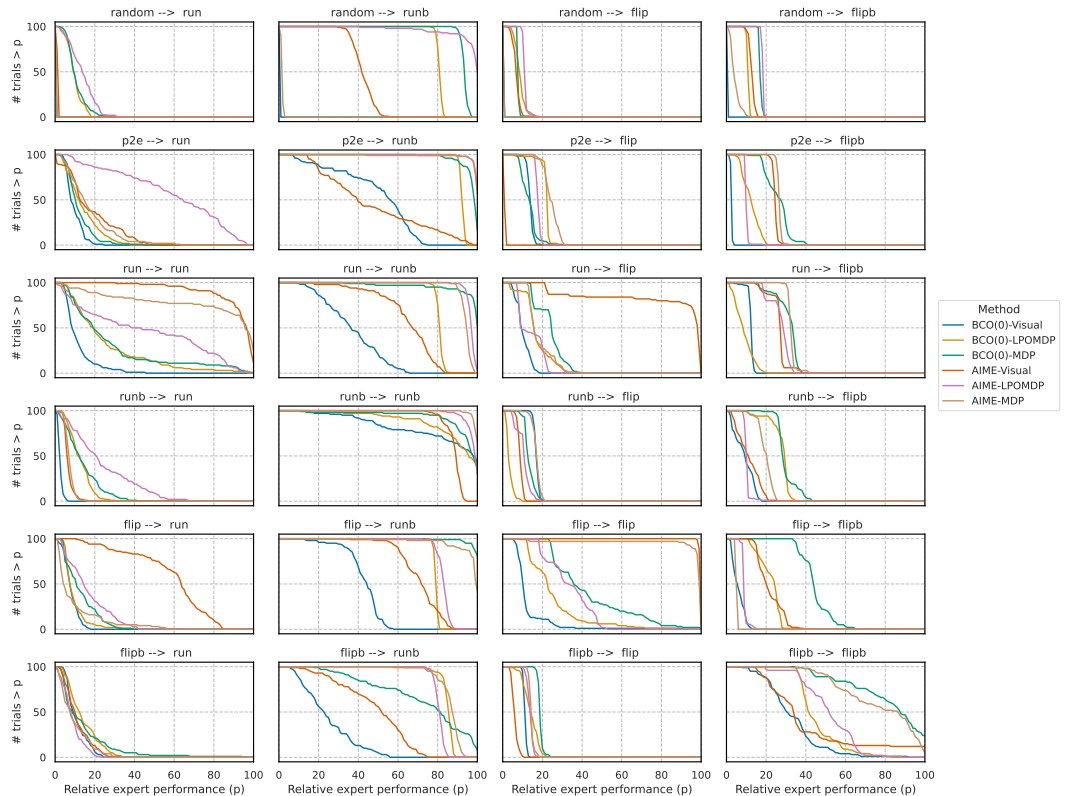

Figure 7: Performance distributions of each method on Cheetah tasks.

show the transfer from the random dataset to the three tasks in the Visual settings which we consider as failure cases. For the stand and walk tasks, none of the metrics are converging. For the run task, we can observe a severe over-fitting starting from the beginning of the training, and the MSE keeps increasing. We conjecture these are all due to the less well-trained world models.

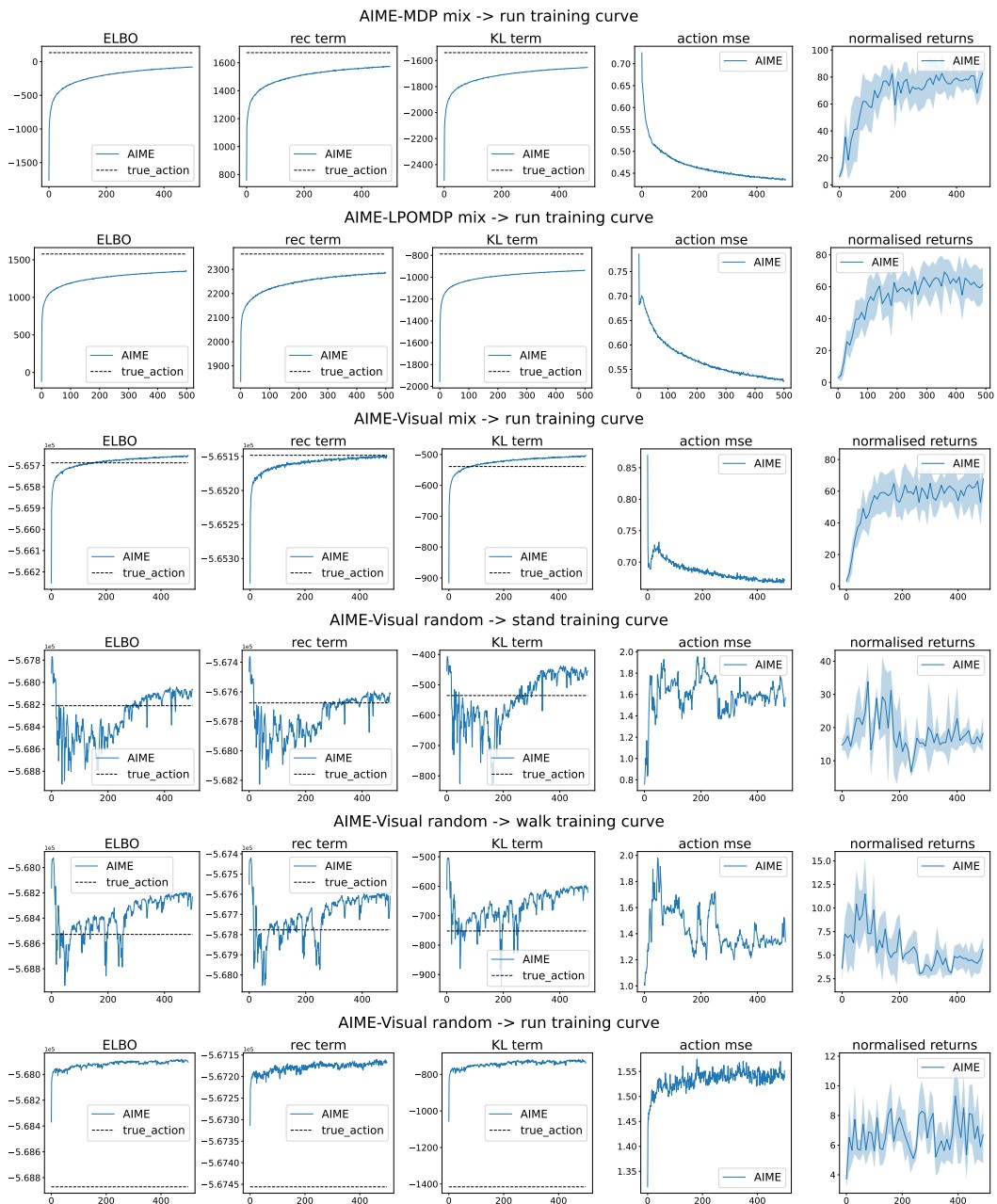

Figure 8: Samples of training curve in phase 2 of AIME. The first three showcase the typical successful training curves, while the remaining three demonstrate the failure cases. The true_action is referring to evaluating the trajectories with the true action sequence.

