# OpenReview forum: "Action Inference by Maximising Evidence: Zero-Shot Imitation from Observation with World Models"
_NeurIPS.cc/2023/Conference — NeurIPS 2023 poster_

### Official Review · Reviewer_MApq · 2023-07-03

**Soundness:** 3 good
**Presentation:** 3 good
**Contribution:** 3 good
**Rating:** 7
**Confidence:** 4

**Summary:**

This paper considers the setting where we have access to a dataset containing states and actions for pre-training and must then learn to solve a task given a new observation-only dataset from a downstream task. The key assumption here is that we have access to this action labelled dataset, and once we have it, we are better off training a world model than simply learning to label the new observation only trajectories with action labels (a la BCO and subsequently VPT). The proposed method is one of the first (as far as I know) to use forward dynamics models to implicitly model the action distribution and provides a nice alternative to IDM approaches for what is becoming an increasingly relevant problem setting. Given the relevance of the topic area, and the fact this exact thing has not been done before, I am voting for the paper to be accepted. My score would be increased if some additional experiments could be conducted, since these particular ones are relatively similar and also low impact in terms of their ambition.

**Strengths:**

The strengths of this work are clear, it is a very relevant problem setting and this method is distinct vs. previous methods like BCO which rely on an IDM. In reality, this paper is essentially "Implicit action learning with forward dynamics models", and that has not been done before as far as I am aware. The method itself is fairly clearly presented, and the experiments are relatively clear with sufficient ablation studies. Finally, it is great to see limitations adequately discussed in the main body, which is surprisingly rare.

**Weaknesses:**

Note that I have voted to accept, the following comments are not red flags but would likely improve the paper, and maybe make it possible to increase to a higher score.
1. The experiments are fairly mundane, and while scientific best practices appear to have been followed, there is a huge gap between what the authors claim to be working towards ("a single foundational world model") and what is actually shown (two DMC environments). It would be fantastic to see an example of this method at larger scale, even if the results are not state-of-the-art and there is only a single seed, for compute reasons. For example, this could be done using the dataset from VPT. The Minecraft images could be resized to make them smaller, and then it would be possible to use the DreamerV3 codebase (which was tested on MineCraft and runs on a single GPU) then see if it is possible to learn from the unlabelled MineCraft videos. If this works, it would drastically increase the impact of the paper, beyond being something mildly interesting for people who care about this specific topic, to something that catches people's eye across the field.
2. It seems slightly fishy to use a world model generated dataset to compare a world model based method and an IDM approach. It is possible there is some bias in the Dreamer or P2E data that makes it easier for a more similar approach to do well. Given how brittle many of these methods are, this could make a difference. Would it be possible to instead consider some open-source benchmarks such as VD4RL (Lu et al), which would make it "fair" across different approaches? For example you could use the random or mixed datasets for the embodied and then expert for the demonstration.
3. All of these experiments are within the same single environment, with 3 different reward functions. There is no variability in terms of the dynamics or observation space. To take a tiny step towards a "foundational world model" could you consider some variation, such as using the distracting control suite or varied dynamics in the simulator? This is also available in VD4RL so could be used there too. My guess is the world model approach would actually do better and it would make the results more interesting.
4. This is a very active area of research so the citations seem light, for example:
- Edwards et al. "Imitating latent policies from observation". ICML 2019
- Seo et al. "Reinforcement learning with action-free pre-training from videos". ICML 2022
- Schmeckpeper et al. "Reinforcement learning with videos: Combining offline observations with interaction". CoRL 2020

**Questions:**

How was the IDM baseline tuned? For example, in the VPT paper they mention different architecture choices made a big difference for performance.

**Limitations:**

Limitations are discussed thoroughly in the main body.

---

> ### Author Rebuttal · Authors · 2023-08-09
>
> > The experiments are fairly mundane. It would be fantastic to see an example of this method at larger scale. For example, this could be done using the dataset from VPT. The Minecraft images could be resized to make them smaller, and then it would be possible to use the DreamerV3 codebase (which was tested on MineCraft and runs on a single GPU) then see if it is possible to learn from the unlabelled MineCraft videos. If this works, it would drastically increase the impact of the paper, beyond being something mildly interesting for people who care about this specific topic, to something that catches people's eye across the field.
>
> We agree doing the MineCraft experiment is a great way to improve the impact of this paper, and we have also thought about that. Actually, we had contacted the authors of DreamerV3 about the possibility to open source the pretrained world model immediately after the paper came out. But the authors said they won't do it due to the extra effort to make their model loadable with the open sourced code base. And if we train the world model ourselves, as stated in the appendix of DreamerV3, it will take about 17 days to train their model. And the VPT labelled dataset is almost as twice large as the final reply buffer of the DreamerV3 agent, so it will probably need a month to train a model and not speak about hyperparameter tuning. As a small lab, we don't have enough resource for an experiment at this scale. But we think this is a great suggestion, and we would like to find ways to try it out in the future.
>
> > It seems slightly fishy to use a world model generated dataset to compare a world model based method and an IDM approach. It is possible there is some bias in the Dreamer or P2E data that makes it easier for a more similar approach to do well. Given how brittle many of these methods are, this could make a difference. Would it be possible to instead consider some open-source benchmarks such as VD4RL (Lu et al), which would make it "fair" across different approaches? For example you could use the random or mixed datasets for the embodied and then expert for the demonstration.
>
> That is a good suggestion. We conduct experiments with the V-D4RL main datasets. Please kindly check the experiment setup and results from the general response.
>
> We can see that the performance of both BCO(0) and AIME is generally low, but AIME still outperform BCO(0) which proves AIME can also handle datasets generated by model-free methods. The low performance is due to more constrained setup of the task, i.e. less amount of embodiment data and less diversity. Except the cheetah-medium_replay having 400 trajectories, the other three datasets provided by VD4RL have only 200 trajectories, which is much less than the 1000 trajectories in our paper. Moreover, it is already shown from Fig. 2 and Fig. 3 that random datasets does not help much about learning a model, and intuitively the medium_replay dataset is better but still does not contain enough information to solve the task.
>
> We would like to point out that due to the limited time in the rebuttal period, these are only preliminary results.
>
> > All of these experiments are within the same single environment, with 3 different reward functions. There is no variability in terms of the dynamics or observation space. To take a tiny step towards a "foundational world model" could you consider some variation, such as using the distracting control suite or varied dynamics in the simulator? This is also available in VD4RL so could be used there too. My guess is the world model approach would actually do better and it would make the results more interesting.
>
> Thanks for you suggestion. We conduct additional experiments with the distracting datasets. Please kindly check the experiment setup and results from the general response.
>
> As we can see from the result, although still outperform the BCO(0) baseline, AIME is largely influenced by the distractions. This behaviour is desired since the world model is trained with reconstruction loss. It is not easy to handle observations with distractions. A potential solution to this problem is to freeze only the dynamic part of the world model and allowing the encoders and the decoders also get finetuned in the second phase. But due to the limited time in the rebuttal, we are unable to test this idea.
>
> > This is a very active area of research so the citations seem light.
>
> Thanks for your suggestion. We will cite these papers in the updated version.
>
> > How was the IDM baseline tuned? For example, in the VPT paper they mention different architecture choices made a big difference for performance.
>
> That is a good point. We are sorry that we forgot to add the implementation details of the BCO baseline to the appendix. In order to make a fair comparison in this paper, the IDM and policy are built by using the same network architecture with the world model. Especially for the visual setting, the IDM uses the same CNN as the world model, and the temporal information is handled by stacking the representation from each frame. Then the action is predicted as a TanhGaussian distribution with an MLP. We did a grid search about the width, depth of the MLP and also the number of stacking frames and didn't find any increase of the performance. We will add these details to the Appendix B for the revised version.
>
> We did not conduct experiments on other architecture design, like using a heavy transformer model to handle longer context length. But from Appendix D.1 of the VPT paper, the most important design choice is using the 3D convolution before the CNN to process each frame. We think that is quite similar to stacking the image directly when you have a short context length and don't have a transformer afterward who needs tokens from each frame. And, according to [1], stacking representations is actually a better design choice than stacking frames.
>
> [1] Shang *et al.*, Reinforcement Learning with Latent Flow, NeurIPS 2021

---

> > ### Comment · Reviewer_MApq · 2023-08-11
> > **Seems good!**
> >
> > Thank you for your response, it seems sensible. I would still like to see something a bit more ambitious for a super high score, but I think this paper should be accepted. The new experiments do provide incremental confidence given they are open source benchmarks vs. author generated settings. Given the other scores are borderline I am willing to support it with a 7.

---

### Official Review · Reviewer_nqaR · 2023-07-03

**Soundness:** 2 fair
**Presentation:** 3 good
**Contribution:** 2 fair
**Rating:** 6
**Confidence:** 4

**Summary:**

This paper presents an imitation learning approach by first training a world model to predict next observations conditioned on (given) actions, and in a second phase training a policy that amortizes action inference by maximizing the likelihood of observations under a dataset of expert demonstrations. The authors compare their method to BCO(0) on the DMC walker and cheetah environments.

**Strengths:**

The paper is well written, the idea is clearly explained and the benchmark seems well executed.

**Weaknesses:**

The experimental results are hard to parse and have some anomalies (see my questions). Also the strongest claims "AIME outperforms the baselines by a large margin, which indicates the strong generalisability of a forward model over an inverse model. We also find that AIME makes substantially better use of exploratory data." are also mainly based on the Walker experiment, but are less outspoken on the Cheetah dataset.

**Questions:**

How is the performance measured? Is this accumulated reward (normalized to the expert performance)?

There are some questioning data points in the evaluations, for instance:
- AIME on Cheetah run->flip is able to learn from Visual, but not at all from LPOMDP/MDP, same for flip->run
- On flipb->run on Cheetah is impossible to learn, but running backwards is
- BCO(0)-MDP seems to outperform AIME-MDP on xxx->flipb
Do you have any insights on these results, whether this is due to the task at hand, the collected dataset, ...?
Moreover, on Cheetah the resulting agent reaches an overall performance of under 50%. Any idea why this performance gap is there?


**Limitations:**

The authors adequately address the limitations in the conclusion.

---

> ### Author Rebuttal · Authors · 2023-08-09
>
> > The experimental results are hard to parse and have some anomalies (see my questions). Also the strongest claims "AIME outperforms the baselines by a large margin, which indicates the strong generalisability of a forward model over an inverse model. We also find that AIME makes substantially better use of exploratory data." are also mainly based on the Walker experiment, but are less outspoken on the Cheetah dataset.
>
> We agree that we did not put enough analysis of the cheetah results. But the conclusion in the paper generally apply to both embodiments, just the improvement on cheetah embodiment is smaller. We will clarify this and add more analysis about the cheetah experiments in the revised version.
>
> > How is the performance measured? Is this accumulated reward (normalized to the expert performance)?
>
> Thanks for bringing this up. Yes, you are right, the performances in all figures are reported based on the normalised accumulated reward. We will add the definition of performance to Sec. 4 in the revised version.
>
> > AIME on Cheetah run->flip is able to learn from Visual, but not at all from LPOMDP/MDP, same for flip->run
>
> We are also confused about this. But we have to point out the RSSM (the dreamer model) is mostly optimised toward visual setting, and we find it harder to tune for other settings. And for the particular settings you mentioned, there is some small similarity between the run and flip tasks, since they all require some subtle movement of the front leg, we conjecture that the change of visual observation can be more pronounced for these subtle movements than the proprioception.
>
> > On flipb->run on Cheetah is impossible to learn, but running backwards is
>
> This is due to the low performance of the run backward expert. In Appendix D, we show the average return of the expert on each task. Run's expert gets a return of $888.65$ while run backward's expert gets only $218.50$. This makes imitating run backward the easiest task among all.
>
> Moreover, we would like to point out that although run and run backward, flip and flip backwards sounds quite similar, due to the asymmetrical structure of the embodiment, the behaviour for solving the tasks can be quite different.
>
> > BCO(0)-MDP seems to outperform AIME-MDP on xxx->flipb Do you have any insights on these results, whether this is due to the task at hand, the collected dataset, ...?
>
> The flips are the hardest tasks in the experiments, since the majority of the time in the expert demonstrations, the cheetah is "flying" in the air and the actions taken there is not relevant for solving the tasks. That leaves only a few actions in the sequence that are actually essential for solving the task. We think this put additional challenge for AIME since it needs to infer a sequence of action, while BCO(0) is operated on point estimation. For example, when the first a few actions cannot output reasonable actions to start the flip, then the later actions will create very noisy gradient since none of them can explain the "flying".
>
> > Moreover, on Cheetah the resulting agent reaches an overall performance of under 50%. Any idea why this performance gap is there?
>
> This is a good question. We think this gap could be the result of different control frequency between the two embodiment. Although it is not well-stated in the literature, with an action repeat of 2 as following other papers, the Walker is running on 20Hz while Cheetah is running on 50Hz. Running in a higher frequency makes each action have smaller influence on the environment, and observations as a consequence. This smaller change makes both the IDM and world model harder to train. There are also works [1, 2] using an action repeat of 4 for the Cheetah environment to get better results, which brings the control frequency to 25Hz, thus closer to the Walker one.
>
> [1] Hafner *et al.*, Learning Latent Dynamics for Planning from Pixels, ICML 2019
>
> [2] Hansen *et al.*, Temporal Difference Learning for Model Predictive Control, ICML 2022

---

> > ### Comment · Reviewer_nqaR · 2023-08-11
> >
> > I thank the authors for their detailed responses on my (and other reviewers') questions. I also appreciate the extra results provided.

---

### Official Review · Reviewer_eCVB · 2023-07-06

**Soundness:** 3 good
**Presentation:** 3 good
**Contribution:** 2 fair
**Rating:** 5
**Confidence:** 3

**Summary:**

This paper presents an algorithm named AIME to learn the world model and apply it to downstream tasks. In the first stage, AIME learns a world model from a dataset with actions to maximize the likelihood via EBLO. While in the second stage, given observation-only demonstrations, AIME optimizes the action sequence to imitate the expert’s behavior. The empirical result shows that AIME outperforms previous methods in DMC tasks.

**Strengths:**

1. Optimizing actions from observation-only trajectory via ELBO is somewhat novel compared to behavior cloning methods.

2. AIME performs better than previous methods even with local and image-based observations.

**Weaknesses:**

1. The major concern is the problem setting of AIME. In my view, more general setting the agent can only get state trajectory in the world-model learning stage while can obtain action-labeled data in the second stage. Then the agent has much more data in training (e.g., human data without actions) and only need a small amount of action-labeled data for fast adaptation. The authors should clarify the significance of the problem settings studied in this paper.

2. Since the world model applies action-labeled data in the first stage, the world model will be related to the policies that generate the dataset. I wonder if the model can handle datasets with a mixture of policies or low-quality policies.



**Questions:**

Why the performance seems to have a very large variance in Figure 3?

---

> ### Author Rebuttal · Authors · 2023-08-09
>
> > The major concern is the problem setting of AIME. In my view, more general setting the agent can only get state trajectory in the world-model learning stage while can obtain action-labeled data in the second stage. Then the agent has much more data in training (e.g., human data without actions) and only need a small amount of action-labeled data for fast adaptation. The authors should clarify the significance of the problem settings studied in this paper.
>
> You raise a good point. We agree that having a large amount of observation-only dataset, like YouTube videos, and then adapt to some action-label datasets is a setting that has attracted a lot of attention in recent years. However, the reversed setting, as we used in this paper, also has a lot of potential. By emphasising embodiment, we require the agent to have a dataset about the embodiment in the first place.
>
> It quite fits for robotics learning [1, 2], where people experiment with the same robot for years and naturally have access to a lot of embodiment data. These years, there are more and larger embodiment datasets getting collected [3] and open sourced [4, 5]. Moreover, it also holds true for well-studied simulator benchmarks where embodiment datasets can be easily access through open sourced benchmarks [6, 7, 8]. Besides, using embodiment data before others is also true for lifelong learning of humans. Thinking about infants who first randomly explore what their body can do before they learn some complex motor skills like walking. Last but not least, learning a model for certain embodiments is much easier than learning a general model for all the embodiments, which we don't even have in nature.
>
> Thus, although the setting that first has embodiment datasets is less popular at the moment, it still has a great potential. We will update the paper to make the motivation of this setting more clear.
>
> > Since the world model applies action-labeled data in the first stage, the world model will be related to the policies that generate the dataset. I wonder if the model can handle datasets with a mixture of policies or low-quality policies.
>
> Yes, the dataset for training the world model matters a lot, and it is also one of the conclusion from the paper. In the experiment section, we train the models on multiple datasets. The random dataset can represent the low-quality policies setting, and all the other datasets are collected by a mixture of policies since they stem from a replay buffer where each trajectory is collected by a different policy, and also the mix dataset contains multiple behaviours. We observe that in general, both AIME and BCO(0) don't work well on the low-quality random dataset since the experience from there hardly explains the observations seen in the demonstrations, i.e. lack of information, and the performance generally improved on p2e dataset where the quality of exploration improves. And AIME works well on the mixture of policies as we showed in other experiments.
>
> > Why the performance seems to have a very large variance in Figure 3?
>
> This is because the methods do well on some tasks but fail on others, when you average them all, you get a very huge variance. We agree it is not an ideal visualization, but we think it is still valuable to report the aggregated result. There are reward profile figures in the appendix E where you can find more details of each test trajectory.
>
> [1] Thrun and Mitchell, Lifelong robot learning, Robotics and Autonomous Systems, 1995
>
> [2] Singh, Transfer of learning by composing solutions of elemental sequential tasks, Machine Learning, 1992
>
> [3] Brohan *et al.*, RT-1: Robotics Transformer for Real-World Control at Scale, arXiv 2212.06817
>
> [4] Dasari *et al.*, RoboNet: Large-Scale Multi-Robot Learning, CoRL 2019
>
> [5] Ebert *et al.*, Bridge Data: Boosting Generalization of Robotic Skills with Cross-Domain Datasets, arXiv 2109.13396
>
> [6] Fu *et al.*, D4RL: Datasets for Deep Data-Driven Reinforcement Learning, arXiv 2004.07219
>
> [7] Qin *et al.*, NeoRL: A Near Real-World Benchmark for Offline Reinforcement Learning, NeurIPS 2022 Datasets and Benchmarks Track
>
> [8] Baker *et al.*, Video PreTraining (VPT): Learning to Act by Watching Unlabeled Online Videos, arXiv 2206.11795

---

> > ### Comment · Reviewer_eCVB · 2023-08-18
> > **Response**
> >
> > Thanks for the response. The limitation of this work in low-quality data needs more research in the future. I will keep the score unchanged.

---

> > > ### Author Response · Authors · 2023-08-18
> > > **Regarding low-quality data**
> > >
> > > We would like to thank you for your reply.
> > >
> > > Regarding the comment on low-quality datasets, we would like to kindly clarify that it will generally influence all the algorithms applied to the problem. In the problem setting, the agent needs to use the knowledge in the embodiment dataset to infer the actions in the demonstration dataset. When the embodiment dataset is in low-quality, meaning it doesn't contain enough knowledge to infer the actions and may make the problem infeasible. To make it more concrete, the Walker-random dataset mainly contains trajectories of the Walker agent laying on the ground, and the Cheetah-random dataset mainly contains trajectories of the Cheetah agent swaying around the starting position. Given this knowledge, it is not possible to fully infer the actions of a complex behaviour like run without additional information. Thus, for any purely data-driven algorithm, the performance of each setup is upper-bounded by the quality of the embodiment dataset, and an algorithm that can utilise the knowledge better can achieve closer results to that upper-bound. In the experiment section, we show AIME is mostly outperforming BCO(0) when using a low-quality random dataset.
> > >
> > > Please let us know if there are any further points of concern or clarification needed.

---

### Official Review · Reviewer_Cs3Z · 2023-07-06

**Soundness:** 2 fair
**Presentation:** 3 good
**Contribution:** 2 fair
**Rating:** 6
**Confidence:** 4

**Summary:**

The paper proposes action inference by maximising evidence as a way for an MBRL to replicate most likely actions using appropriate world models. The algorithm has two phases: 1) Learn the world model based on a replay buffer, and 2) imitate the expert's behaviour by inferring the policy that maximizes the evidence of the demonstration under the policy and world model. Experimental results on the Walker and Cheetah embodiments of the DeepMind Control Suite demonstrate that this zero-shot imitation performance outperforms the current state-of-the-art approaches.

**Strengths:**

-	The paper addresses a major issue in deep reinforcement learning (DRL), namely sample inefficiency. By suggesting a method that can harness observational data, the authors propose a way to improve the sample efficiency of DRL agents.
- The paper introduces a new method, Action Inference by Maximising Evidence (AIME), for imitation learning. This method is designed to mimic the human ability to learn quickly from observation, which is an interesting contribution.
- AIME's two-phase learning process, involving the creation of a world model and then using it for imitation learning, is a unique approach. This process allows the agent to understand its own body and the likely actions that led to observed behaviors, which is a crucial aspect of learning.
-  The method is capable of "zero-shot" learning, meaning it does not require further training for the world model or online interactions with the environment after being given the demonstration. This is a significant advantage in terms of efficiency and practicality.
- The authors provide empirical validation of their method on the Walker and Cheetah embodiments of the DeepMind Control Suite. They demonstrate that their method outperforms state-of-the-art baselines, which strengthens the credibility of their approach.

**Weaknesses:**

-	The paper validates the AIME method using the Walker and Cheetah embodiments of the DeepMind Control Suite, which are simulated environments. It's unclear how well the method would perform in real-world scenarios, where conditions can be more complex and unpredictable.
-	The AIME method assumes that the agent can learn a perfect world model from its past experiences. This may not always be possible due to the complexity and unpredictability of many environments. The performance of the method could be affected if the world model is not accurate.
-	The effectiveness of the AIME method may heavily depend on the quality of the observation-only demonstrations provided. If these demonstrations are not representative of the task at hand, or if they are of poor quality, the performance of the method could be significantly affected.
-	The paper does not discuss the computational complexity of the AIME method or its scalability to larger and more complex tasks. If the method is computationally intensive, it may not be practical for use in real-time applications or on larger scales.

**Questions:**

-	Does the world model training in Phase 1 have to converge because imitation learning can happen? Is this primarily for changes in the task? E.g., going from walking to hopping but with the same agent in the same environment?
-	What if the space of actions change between phase 1 and phase 2? Will AIME still work? It doesn’t seem like it.
-	Can action trajectories be learnt directly instead of one step action policies?
-	How sensitive is the AIME method to the quality and diversity of the observation-only demonstrations provided? What happens if the demonstrations are not representative of the task at hand or are of poor quality?
-	What is the computational complexity of the AIME method? How well does it scale to larger and more complex tasks?
-	does the AIME method handle situations where the world model learned from past experiences is not accurate or complete? What are the implications if the world model is imperfect?
-	How well does the AIME method perform in terms of transfer learning? Can the world model learned in one context be effectively applied to another for the policy?
-	How robust is the AIME method to changes in the environment or task? Can it adapt to new situations without requiring additional training?

**Limitations:**

The authors point out the following limitations that limit the scope of the current results:

- The AIME method performs well with visual input, but there is a significant performance gap when compared to the LPOMDP setting where low-dimensional signals are observed. This is attributed to the loss surface of the pixel reconstruction loss not being smooth enough to allow the gradient method to find an equally good solution.
- The study only considers the simplest setting where both the embodiment and sensor layout are fixed across tasks. This is a limitation as humans observe others in a third-person perspective and can imitate animals whose bodies are not similar to humans. Relaxing these assumptions could allow for transfer across different embodiments and even directly from human videos.
- For some tasks, even humans cannot achieve zero-shot imitation by only watching others. This could be due to the task's complexity or completely unfamiliar skills. Even with proper instruction, humans still need to practice in the environment and learn something new to solve some tasks. This suggests the need for an online learning phase as an extension to the AIME framework.

---

> ### Author Rebuttal · Authors · 2023-08-09
>
> > Does the world model training in Phase 1 have to converge because imitation learning can happen? Is this primarily for changes in the task? E.g., going from walking to hopping but with the same agent in the same environment?
>
> No, it is not necessary to train the model until converge to enable imitation. It emerges gradually during training. From the results shown in Fig. 1 in the general response, the imitation ability is established in very early phase of the training process long before convergence.
>
> > What if the space of actions change between phase 1 and phase 2? Will AIME still work? It doesn’t seem like it.
>
> In general, we do not care about what action space is used to collect the demonstrations for phase 2, since we only need the observations. AIME will derive a policy in the original action space that the world model is trained on during phase 1. This is actually a benefit of our method, since it allows you to have different action space for the agent and the demonstrator. For example, if you want to demonstrate a task on a robot arm, the low-level action space is not intuitive for a human operator. In our setup we can easily use another more intuitive interface like smart phones [1] to do the demonstration and let the agent to imitate on the low-level action space. But if you are referring to completely changing the action space also in deployment, which is not likely to happen in real life, it will require an extra function to map the action from the old action space to the new action space.
>
> > Can action trajectories be learnt directly instead of one step action policies?
>
> Yes, they can. As we show in the control derivation from Sec. 3.2, we can infer the action trajectories directly by a planning algorithm. But it needs to be done individually for each trajectory. That is why we introduce amortised inference to improve the efficiency. And also one can define the amortised inference model in many different forms, for example $\pi(a_{t:t+T}|s_t)$. We use the one step action policy form to keep it simple and comparable to the baselines.
>
> > How sensitive is the AIME method to the quality and diversity of the observation-only demonstrations provided? What happens if the demonstrations are not representative of the task at hand or are of poor quality?
>
> We have to clarify that, in the setting of imitation, the task is defined by the demonstrations. We aim to replicate the behaviour of the demonstration rather than solving the original task that the demonstrator is trying to solve. This can be viewed as an alternative way of defining the task, i.e. by showing how the task is done, rather than defining a reward function.
>
> About the diversity of the demonstrations, we would like to point you to Fig. 4 of the paper, where we limited the number of demonstrations, which also limited the diversity. We can see this indeed has an effect on the performance, but AIME is less sensitive to this than the BCO(0) baseline.
>
> > What is the computational complexity of the AIME method? How well does it scale to larger and more complex tasks?
>
> For the training complexity, in Appendix A, we show phase 1 requires 10 - 20 hours while phase 2 requires 5 - 10 hours on an old 1080ti GPU. For solving multiple tasks, you only need to run phase 1 once and run phase 2 multiple times for different tasks. Moreover, we would like to mention that 5 -10 hours training time is for training 500 epochs, but from Fig. 8 of Appendix E, normally AIME does not need that long for converging. This makes AIME require even less compute.
>
> For the inference complexity, we are training a RSSM world model from the Dreamer papers. So we can do real-time inference whenever their method can. In a recent paper [2], they successfully run the model to control four different robots in real-time. Thus, our method is also applicable for the real-time scenarios.
>
> For the scalability, we also run experiments on the mix dataset, where the dataset is 3 times larger than other datasets in this paper. The improved results suggest AIME can handle a larger scale of data.
>
> > Does the AIME method handle situations where the world model learned from past experiences is not accurate or complete? What are the implications if the world model is imperfect?
>
> We didn't assume the world model learnt on the first phase is perfect. And actually, all the pretrained world models considered in the experiments are not perfect, since the training dataset did not cover all the dynamic of the embodiment.
>
> When the world model is imperfect, it could degenerate the imitation performance, like in the convergence experiment, or in the worst case leads to failure like divergence. We show a few of these failure cases in the Appendix E.
>
> > How well does the AIME method perform in terms of transfer learning? Can the world model learned in one context be effectively applied to another for the policy?
>
> This transfer ability is exactly what we show with AIME. In the majority of the experiments, we do cross task transfer. For example, we learn a world model on the stand task, then imitate the policy of the run task. The results showcase the strong transfer ability of the world model.
>
> > How robust is the AIME method to changes in the environment or task? Can it adapt to new situations without requiring additional training?
>
> In this paper, we mainly assume the same environment in both phases. In general, the robustness to environment change can be injected through diverse training dataset or domain-knowledge-based data augmentation. In term of tasks, that is exactly what we show in the paper. The pretrained world model exhibit the zero-shot ability for imitation, without further environment interactions to finetune the model.
>
> [1] Mandlekar *et al.,* Scaling Robot Supervision to Hundreds of Hours with RoboTurk: Robotic Manipulation Dataset through Human Reasoning and Dexterity, IROS 2019
>
> [2] Wu *et al.*, DayDreamer: World Models for Physical Robot Learning, CoRL 2022

---

> > ### Comment · Reviewer_Cs3Z · 2023-08-21
> >
> > Thank you for your response - I am convinced about the technical contribution of the paper and happy to support acceptance. I will be increasing my score by 1 point.

---

### Author Rebuttal · Authors · 2023-08-09

We thank all reviewers for their time and insightful feedbacks.

We conduct three new experiments suggested by the reviewers, the results are provided in the pdf:

**We evaluate multiple checkpoints during the course of the world model pretraining to address the convergence concerns from reviewer Cs3Z. To be specify, we retrain the model on walker-mix dataset for 2000 epochs (prolong from 1000 epochs in the paper) and save a checkpoint every 100 epochs. Then, all the 20 saved models are evaluated by imitating the run policy.**

From the results shown in Fig. 1, the imitation ability is established in very early phase of the training process long before converge.

**We conduct experiments on the VD4RL [1] main datasets, as suggested by reviewer MApq, to prove that AIME can also work with dataset generated by model-free methods. We use their random and medium_replay datasets from Walker and Cheetah as embodiment datasets. Besides, we also merge these two datasets to form a mix dataset. The models are trained on the three datasets for each embodiment. Then we treat the expert datasets in the benchmark as demonstration dataset.**

We can see that the performance of both BCO(0) and AIME is generally low, but AIME still outperform BCO(0) which proves AIME can also handle datasets generated by model-free methods. The low performance is due to more constrained setup of the task, i.e. less amount of embodiment data and less diversity. Except the cheetah-medium_replay having 400 trajectories, the other three datasets provided by VD4RL have only 200 trajectories, which is much less than the 1000 trajectories in our paper. Moreover, it is already shown from Fig. 2 and Fig. 3 that random datasets does not help much about learning a model, and intuitively the medium_replay dataset is better but still does not contain enough information to solve the task.

**We conduct experiments on the VD4RL [1] distracting datasets, as suggested by reviewer MApq, to test the performance of AIME on distracting datasets. For the walker embodiment, the benchmark provides random datasets with a distraction level of easy, medium, and hard. We also merge these three level to a mix dataset. Moreover, we also merge this mix dataset with the mix dataset in the second experiment to form a total_mix dataset. We treat these five datasets as the embodiment dataset and the expert dataset as the demonstration dataset. For the cheetah embodiment, the benchmark provides medium and expert datasets with a distraction level of easy, medium, and hard. We subsample the medium datasets to get 200 trajectories with each level, then merge that with the mix dataset in the second experiment to form a total_mix dataset. Then the algorithms are using this total_mix dataset as the embodiment dataset and the expert dataset as the demonstration dataset.**

As we can see from the result, although still outperform the BCO(0) baseline, AIME is largely influence by the distractions. This behaviour is desired since the world model is trained with reconstruction loss. It is not easy to handle observations with distractions. A potential solution to this problem is to freeze only the dynamic part of the world model and allowing the encoders and the decoders also get finetuned in the second phase. But due to the limited time in the rebuttal, we are unable to test this idea.

We would like to mentioned that due to the limit time during rebuttal, we only run AIME and BCO(0) baselines with the default parameters we used in the paper. Further hyper-parameter tuning could achieve a better result.

For individual questions raised by each reviewer, please find the responses below.

[1] Lu *et al.*, Challenges and Opportunities in Offline Reinforcement Learning from Visual Observations, TMLR 2023

---

### Decision · Program_Chairs · 2023-09-21

**Decision:**

Accept (poster)

**Comment:**

Reviewers unanimously vote to accept the paper. The paper presents a methd for zero-shot imitation wherein the agent first learns a world model (i.e., forward dynamics model) and then given demonstrations of a target task uses its world model to infer actions. In the second stage, no finetuning or action information is required -- the demonstrations simply consist of a sequence of observations. Results are presented on a few locomotion tasks. The diversity of the tasks are a concern amongst the reviewer and therefore their ratings are lukewarm.

While I am going with the reviewers and recommending acceptance, I encourage the authors to include the following discussion in the introduction:

- There is a rich history in robotics where the "robot" first learns some basic skills and then uses them to interpret video/observation only demonstrations (see [1]-[4]) -- including them / discussing them is important to set the context for the work. Similarly, consider including other papers that the reviewers have pointed out, e.g., MApq pointed out:
Edwards et al. "Imitating latent policies from observation". ICML 2019
Seo et al. "Reinforcement learning with action-free pre-training from videos". ICML 2022
Schmeckpeper et al. "Reinforcement learning with videos: Combining offline observations with interaction". CoRL 2020

- Inverse Dynamics Models (IDM) have been used exactly in this setting, and the authors cite and compare against it. However, more advanced IDM based approaches such as in [6] can easily be applied in locomotion tasks -- however authors argue they can't be in Lines 171-72. Even in locomotion tasks, given a pair of frames $o_t, o_{t+1}$, the frame $o_{t+1}$ can be considered to be the goal and can the trained goal-conditioned models can be applied. Please either provide a comparison with [6] or discuss in detail why it is not applicable. Further, the discussion of [5],[6] and other IDM based methods should be included in the introduction.

The strength of this in my opinion is not that the agent's embodiment is learned first -- IDM approaches also do that. It is really that the authors are learning forward models -- and therefore its key to compare against strong implementations of IDM such as in [6].

[1] Yezhou Yang, Yi Li, Cornelia Fermuller, and Yiannis Aloimonos. Robot learning manipulation action plans by ”watching” unconstrained videos from the world wide web. In AAAI, 2015.
[2] Cynthia Breazeal and Brian Scassellati. Robots that imitate humans. Trends in cognitive sciences, 2002.
[3] Yasuo Kuniyoshi, Masayuki Inaba, and Hirockika Inoue. Teaching by showing: Generating robot programs by visual observation of human performance. In International Symposium on Industrial Robots, 1989.
[4] Yasuo Kuniyoshi, Masayuki Inaba, and Hirochika Inoue. Learning by watching: Extracting reusable task knowledge from visual observation of human performance. IEEE Transactions on Robotics and Automation, 1994
[5] Ashvin Nair, Dian Chen, Pulkit Agrawal, Phillip Isola, Pieter Abbeel, Jitendra Malik, and Sergey Levine. Combining self-supervised learning and imitation for vision-based rope manipulation. ICRA, 2017.
[6] Pathak, Deepak, Parsa Mahmoudieh, Guanghao Luo, Pulkit Agrawal, Dian Chen, Yide Shentu, Evan Shelhamer, Jitendra Malik, Alexei A. Efros, and Trevor Darrell. "Zero-shot visual imitation." ICLR 2018